# An Evaluation Index System of China's Development Level of Ecological Civilization

**Xiaotian Wang** [1,2] and **Xingpeng Chen** [1,2,*]

1   College of Earth and Environmental Sciences, Lanzhou University, Lanzhou 730000, China; wangxt17@lzu.edu.cn
2   Research Institute for Circular Economy in Western China, Lanzhou University, Lanzhou 730000, China
*   Correspondence: chenxp@lzu.edu.cn

**Abstract:** Ecological civilization, a word with distinctively Chinese characteristics, will be the key strategy to address a series of problems in China's economic transformation in the future. This study elaborated the concept and connotation of an ecological civilization from both narrow and broad perspectives, and established an evaluation index system with 26 specific indicators. These indicators were based on concepts to evaluate the development level of ecological civilization in mainland China and its 31 provinces (excluding Hong Kong, Macao, and Taiwan), autonomous regions, and municipalities from 2004 to 2016. The results reveal that China has achieved a transition in the development level of ecological civilization from low to intermediate as a whole, however, even rapid economic growth at the expense of the ecological environment cannot enable China to get rid of the fact that its social and economic development is lagging behind developed countries and regions. China and its various provinces, autonomous regions, and municipalities should gradually reduce their own deficiencies in the process of ecological civilization construction, under the premise of maintaining their own advantages, so as to achieve sustainable development and accelerate the construction of an ecological civilization.

**Keywords:** ecological civilization; analytic hierarchy process; entropy weight; comprehensive evaluation index; ecological environment; economic society

## 1. Introduction

With four decades of reform and opening up, China's international status and influence have increased with time. However, China's basic national conditions—China is in the primary stage of socialism and will remain so for a long time to come—means it has to carry out its national economic construction in an extensive mode of economic growth with high pollution, high energy consumption, and a development pattern of "grow first, clean up later". Such a pattern has led China to face great pressures, both in the processes of industrial transforming and upgrading, and of promoting the coordinated development of social economy [1]. As sustainable development and green development concepts have gradually become inveterate, the government and its people have paid more and more attention to the notion of an ecological civilization based on these concepts. China's construction towards an ecological civilization was initially proposed at the 16th National Congress of Communist Party of China (CPC) and was extended at the 17th and 18th National Congress of CPC. Then, at the 19th National Congress of CPC, China further theorized the general guidelines for solving problems in ecological civilization development. This indicates that the ecological civilization theory is becoming increasingly sophisticated [2].

Ecological civilization has harmony between man and nature as its core value orientation. This means it requires human beings to actively remold the natural world on the premise of following

objective laws of nature and fully reflecting the environmental, resource, and economic values of the whole ecosystem. Thus, the ecosystem can achieve an equilibrium conducive to the sustainable development of human society [3–5]. Because there are different natural environments, economic conditions, and social conditions in different regions, regional development disparities normally exist in stages of development [6–8]. These disparities directly lead regions to pursue various alternatives in construction. Therefore, only by combining local development conditions and evaluating the development phase of each region can we clarify the development ideas of an ecological civilization, thereby accelerating development processes.

At present, studies on evaluating the development of an ecological civilization in China have mainly adopted indexing system methodologies. The earliest system was "the index system of ecological county, ecological city, ecological province construction" introduced by the Ministry of Environmental Protection (MeP) in 2007. Subsequently, the MeP and the general office of the state council published a series of indicators, including Indicators of National Ecological Civilization Construction Demonstration zones, Indicators of National Ecological Civilization Construction Demonstration County and City, and Methods for Evaluation and Assessment of Ecological Civilization Construction Goals [9]. However, these indexing systems focus on local planning targets, which are different from the emphasis on research and evaluation of ecological civilization development levels. In order to establish more scientific and accurate indexing systems, many academic researchers joined the ranks of these studies. The earliest one of these indexing systems appeared in a paper published by Yang on the ecological civilization of various provinces, autonomous regions, and municipalities of mainland China in 2009, which established an evaluation index system by taking ecological footprints per capita as a single indicator [10]. Thereafter, research on an indexing system for evaluating ecological civilization development was in full swing. Indexing systems after this have become more and more complex and have integrated other relevant aspects such as material, spiritual, and political civilizations [11]. Others have included an index system from the perspective of regional gaps between China's environmentally problematic regions [12], an index system based on strong sustainable perspectives [13], an index system that combined the Coordinated Development Degree [14], an index system that combined Pressure-State-Impact-Response (PSIR) Model and Structural Equation Modeling (SEM) [15], etc. However, the current evaluation of China's ecological development is mainly based on provincial- or smaller-scale studies, instead of a nationwide scale, and the standards for evaluating the development level of an ecological civilization cannot be unified.

By referring and combining research progres made in existing ecological civilization indexing systems, this study first selected evaluation indicators according to an ecological civilization connotation. Then, by using the analytic hierarchy process (AHP) and entropy weight methods, we determined the weight of each index and established system to evaluate ecological civilization development. Finally, we utilized our indexing system to define a comprehensive index for an ecological civilization. Furthermore, the development types and stages of ecological civilization in mainland China and its provinces, autonomous regions, and municipalities were classified by calculating the numerical results of all indexes, and the patterns of spatial and temporal ecological civilization development were revealed. We also expect to provide theoretical basis and scientific guidance for further construction of an ecological civilization in China.

## 2. Methods

### 2.1. New Concept on the Connotation of Ecological Civilization

An accurate grasp of the concept of ecological civilization and an in-depth understanding of its connotation are the premise and primary basis for the evaluation of the development level of ecological civilization. At present, although a relatively consistent opinion has been formed on the concept of ecological civilization, generally, the discussion on its precise definition and specific connotation continues [16–19]. The connotation of an ecological civilization, we understand, should not stop at

simply protecting the natural environment, but should gradually rise to the height of coordinated development of nature, economy, and society [20,21]. Combined with the existing discussion on the connotation of ecological civilization, we believe that ecological civilization is born out of industrial civilization, and that when industrial civilization develops to a certain stage, the ecological notion will continue to seep into all the subsystems of industrial civilization, such as material, political, and spiritual civilizations. We also believe that these subsystems will experience an ecological transformation, which will exist in the process of negating and affirming industrial civilization itself. We finally believe that human civilization will eventually embrace the evolution from industrial civilization towards a more advanced stage of civilization, and this new advanced stage of human civilization will be known as an ecological civilization. That is to say, for the moment, ecological civilization is just an important notion in industrial civilization, but when it grows stronger and is rooted in various aspects of our social existence, viz., when the material, spiritual, and political civilizations under the current form translate into ecological–material, ecological–spiritual, and ecological–political civilizations, respectively, ecological civilization will replace industrial civilization as a new development stage of human civilization.

### 2.2. Construction of the Comprehensive Evaluation Index of Ecological Civilization

### 2.2.1. Index Selection and the Construction of the Evaluation index System

The index of ecological civilization we chose was based on the concept and connotation of an ecological civilization, and construction of the whole indexing system followed the principles of scientific soundness, comprehensiveness, forward-looking, comparability, and maneuverability [22,23]. According to the general thought of "a set of indicators adjust under the premise of adjusting measures to local conditions", we selected unified evaluation indicators across the country, which were mutually independent, so as to improve the accuracy of our evaluation results and enable the whole indexing system to comprehensively reflect the overall level of regional ecological civilization construction [8,24–26]. Our evaluation index system was as follows:

Target layer A: Evaluate the index systems of ecological civilization development.

Criterion layer B: Include the ecological environment and economic society, which are two core aspects of ecological civilization.

Sub-criterion layer C: We set several sub-criterion layers under each criterion layer. The ecological environment layer was divided into four sub-criterion layers: national territory, ecological pressure, environmental governance, and green residence, which all aimed at evaluating the regional natural conditions and the effectiveness of ecological environmental protection. The economic society aspect included resource conservation, social development, national lives, population quality, and ecological construction, which were all aimed at evaluating regional social and economic development from the perspective of a circular economy and social sustainable development.

Index layer D: Each sub-criterion layer had 2–4 specific indicators. The selected indicators could reflect the level of regional ecological civilization construction and the degree of meeting the connotation and requirements of ecological civilization, as shown in Table 1.

In our evaluation index system, index selection not only combined the advantages of previous studies, but took the scientific and comprehensive evaluation of the development level of regional ecological civilization as its first priority. The relationship between ecological environment and economic society is by no means separate from each other. For example, resource conservation is required for socially and economically sustainable development as well as for ecological and environmental protection because good social and economic development requires us to provide more green residence. We took this into account when we selected indicators. Compared with the existing studies, we selected the indicators by fully considering the balance and relevance between the two basic dimensions of ecological civilization development—ecological environment and economic society—so as to improve the comprehensiveness of the evaluation index system and allow our system have unique advantages in the evaluation of ecological civilization.

**Table 1.** Evaluation index system of ecological civilization (ECI) development.

| Target Layer A | Criterion Layer B | Sub-Criterion Layer C | Index Layer D | Unit |
|---|---|---|---|---|
| Evaluation index system of ecological civilization development (A) | Ecological environment (B1) | National territory (C1) | Forest coverage rate (D1) | % |
| | | | Percentage of national reserves in the region (D2) | % |
| | | | Per capita water resources (D3) | $m^3$/person |
| | | Ecological pressure (C2) | Chemical Oxygen Demand (COD) emissions intensity (D4) | kg/10,000 yuan |
| | | | $SO_2$ emissions intensity (D5) | kg/10,000 yuan |
| | | | Intensity of chemical fertilizer application (D6) Environmental emergencies (D7) | kg/ha. time |
| | | Environmental governance (C3) | Comprehensively utilized rate of common industrial solid wastes (D8) | % |
| | | | Treatment rate of consumption waste (D9) | % |
| | | Green residence (C4) | Proportion of days of air quality equal to or above grade II in the whole year in provincial capital-level city (D10) | % |
| | | | Per capita park green areas in cities (D11) | $m^3$/person |
| | Economic society (B2) | Resource conservation (C5) | Energy consumption intensity (D12) | 10,000 tons of SCE/thousand yuan |
| | | | Water consumption intensity (D13) | $m^3$/10,000 yuan |
| | | | Per land GDP (D14) | 100 million yuan/km$^2$ |
| | | Social development (C6) | Urbanization rate (D15) | % |
| | | | Proportion of tertiary industry (D16) | % |
| | | National lives (C7) | Per capita annual disposable income of urban households (D17) | yuan |
| | | | Per capita annual net income of rural households (D18) | yuan |
| | | | Income ratio of urban and rural residents (D19) | — |
| | | | Engel's Coefficient (D20) | % |
| | | Population quality (C8) | Per capita educational years (D21) | year |
| | | | Illiteracy rate (D22) | % |
| | | | Population life expectancy (D23) | year |
| | | Ecological construction (C9) | Proportion of environmental pollution investment (D24) | % |
| | | | Proportion of R&D expenditure (D25) | % |
| | | | Proportion of Nationally Designated Eco-Demonstration Region/Eco-County (City, District) (D26) | % |

On an ecological environment level (Criterion layer B1), we chose national territory, ecological pressure, environmental governance, and green residence as the four sub-criterion layers. These could not only directly reflect the quality of regional ecological environment, but reflect whether the local ecological environment protection work was conducive to sustainable development of human beings. On an economic society level (Criterion layer B2), although some economic-only measurements (like per land GDP, etc.) have been criticized by many scholars because of their limitations in explaining the level of economic development, they were still irreplaceable at present. So we combined per land GDP, urbanization rate, and proportion of tertiary industry as the indicators of social development. And we took social development, resource conservation, national lives, population quality, and ecological construction as the five aspects of economic society investigation to make up for the defects that may be caused by economic-only indicators in the evaluation of economic society.

### 2.2.2. Index Data Processing

Different types of data could not be directly compared because they had multiple dimensions and magnitudes. Only by standardizing the data and ensuring equal status of all indicators could the data of all indicators be comparable. Therefore, each index should be firstly standardized according to its nature. In this study, all indicators were divided into positive indicators and negative indicators, and the min-max normalization was selected for data standardization.

The formula for the standardized method for the positive indicators is:

$$y_{ij} = \frac{x_{ij} - x_{min}}{x_{max} - x_{min}} \ (i = 1, 2, \ldots n; j = 1, 2, \ldots m). \tag{1}$$

The formula for the standardized method for the negative indicators is:

$$y_{ij} = \frac{x_{max} - x_{ij}}{x_{max} - x_{min}} \ (i = 1, 2, \ldots n; j = 1, 2, \ldots m). \tag{2}$$

Through repeated comparative studies of various methods, we adopted the integration of subjective and objective empowerment to determine the index weight. By choosing the analytic hierarchy process (AHP) method, we first convened 20 relevant experts and held an expert consultation meeting to score for the relative importance of the indicators in sub-criteria layer C and index layer D in order to determine the index weight. Then, we corrected the result by combing a literature review with the entropy weight method, thereby obtaining the ultima index weight. As a note, we considered ecological environment and economic society to be equally important in the construction of ecological civilization, and the weights of the two criteria layers B1 and B2 were judged equivalent as well. The attribute and specific weights of indicators were as shown in Table 2.

### 2.3. Comprehensive Evaluation Index of Ecological Civilization Development

To evaluate the development level of ecological civilization in various regions accurately, we used the comprehensive index method to calculate the scores of various indicators in Index layer D based on the weight of each index and the standardized data. We totaled the scores of each index to obtain ecological comprehensive evaluation indexes of national and provincial ecological civilization (ECI). The calculation method was as follows:

$$ECI = 100 \sum_{i=1}^{26} y_i w_i \tag{3}$$

where $y_i$ is index No. i of index layer D and $w_i$ is the weight of the index No. i. The result of ECI reflected the degree to which an area met the requirements of ecological civilization under its current conditions of human civilization. Then, the level of ecological civilization development was graded according to the ECI score, as shown in Table 3.

**Table 2.** The attributes and specific weights of indicators.

| Index Layer D | Attribute | Weight |
|---|---|---|
| Forest coverage rate (D1) | + | 0.043 |
| Percentage of national reserves in the region (D2) | + | 0.033 |
| Per capita water resources (D3) | + | 0.074 |
| COD emissions intensity (D4) | - | 0.033 |
| $SO_2$ emissions intensity (D5) | - | 0.036 |
| Intensity of chemical fertilizer application (D6) | - | 0.029 |
| Environmental emergencies (D7) | - | 0.053 |
| Comprehensively utilized rate of common industrial solid wastes (D8) | + | 0.046 |
| Treatment rate of consumption waste (D9) | + | 0.054 |
| Proportion of days of air quality equal to or above grade II in the whole year in provincial capital-level city (D10) | + | 0.043 |
| Per capita park green areas in cities (D11) | + | 0.057 |
| energy consumption intensity (D12) | - | 0.039 |
| water consumption intensity (D13) | - | 0.050 |
| Per land GDP (D14) | + | 0.061 |
| urbanization rate (D15) | + | 0.047 |
| Proportion of tertiary industry (D16) | + | 0.024 |
| Per capita annual disposable income of urban households (D17) | + | 0.040 |
| Per capita annual net income of rural households (D18) | + | 0.044 |
| Income ratio of urban and rural residents (D19) | - | 0.048 |
| Engel's Coefficient (D20) | - | 0.013 |
| Per capita educational years (D21) | + | 0.026 |
| Illiteracy rate (D22) | - | 0.019 |
| Population life expectancy (D23) | + | 0.021 |
| Proportion of environmental pollution investment (D24) | + | 0.010 |
| Proportion of R&D expenditure (D25) | + | 0.010 |
| Proportion of Nationally Designated Eco-Demonstration Region/Eco-County (City, District) (D26) | + | 0.015 |

**Table 3.** Stage of ecological civilization development.

| ECI score | <30 | 30–40 | 40–50 | 50–60 | 60–70 | 70–80 | ≥80 |
|---|---|---|---|---|---|---|---|
| ecological civilization development | extremely low | low | comparatively low | intermediate | comparatively high | high | extremely high |

## 3. Results and Discussion

### 3.1. Overall Situation of Ecological Civilization Development in Mainland China

Based on the calculation results of ecological environment and the economic society evaluation score from 2004 to 2016, we further obtained an ECI score and determined the development stage of ecological civilization. The raw data of each index were from the China statistical yearbook, China energy statistical yearbook, China statistical yearbook on environment, educational statistical yearbook of China, China statistical yearbook on science and technology, statistical yearbook and statistical bulletin of provinces, autonomous regions and municipalities, and internet-related resource data from 2004 to 2016 [27–31]. Some economic statistics such as GDP were calculated on the basis of the constant price in 2000. Evaluation results are shown in Figure 1 and Table 4.

Table 4 shows that the ECI score in mainland China as a whole maintained an upward momentum from 2004 to 2016, and the development stage of ecological civilization turned from a low stage in 2004 to an intermediate stage in 2016. The ecological environment score was constantly higher than that of economic society. While compared with the sustainable growth of the economic society score, the ecological environment score increased year by year from 2004 to 2010, then fluctuated within a range of 54 to 60. From the perspective of the development stage of ecological civilization, 2007 and 2012 were taken as two junctures of time, which were the stages of transition from low to comparatively low,



and from comparatively low to intermediate, respectively (although China entered the intermediate development stage of ecological civilization for the first time in 2010, it was not until 2012 that the development stage of ecological civilization was maintained steady at intermediate stage). We saw that China really made great progress in ecological civilization construction in recent years. However, China was still in the intermediate development stage of ecological civilization until 2016, and its ECI score even fluctuated in recent years. This demonstrated that China should close the gap between its ecological civilization development level and that of developed countries and regions, and it should continue to strengthen ecological civilization construction comprehensively in the future [32].

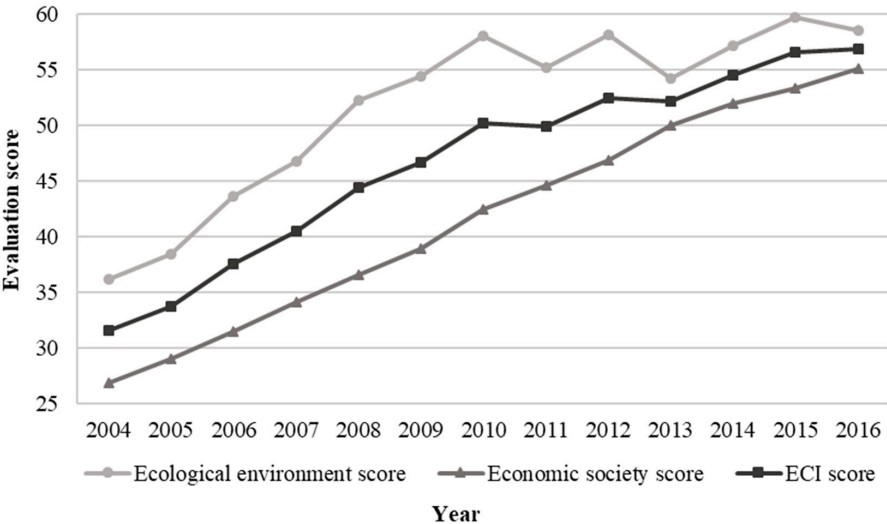

**Figure 1.** Evaluation results of ecological environment, economic society, and ECI.

**Table 4.** Development stage of ecological civilization in mainland China.

| Year | ECI Score | Development Stage of Ecological Civilization |
|------|-----------|----------------------------------------------|
| 2004 | 31.55 | Low |
| 2005 | 33.75 | Low |
| 2006 | 37.57 | Low |
| 2007 | 40.49 | comparatively low |
| 2008 | 44.39 | comparatively low |
| 2009 | 46.65 | comparatively low |
| 2010 | 50.24 | intermediate |
| 2011 | 49.93 | comparatively low |
| 2012 | 52.50 | intermediate |
| 2013 | 52.13 | intermediate |
| 2014 | 54.54 | intermediate |
| 2015 | 56.56 | intermediate |
| 2016 | 56.84 | intermediate |

The core value of ecological civilization is the harmonious coexistence between human and nature, and it requires ecological environment and social economy to advance in a coordinating way [33]. Any practice that focuses on one side while ignoring the other will ultimately be detrimental to the sustainable development of ecological civilization [34–36]. Although evaluation results indicated that China's overall economic and social development lagged behind resource and environmental conditions in the same period, it did not mean that China should put more emphasis on economic development than on protection of ecological environment. China is a country with vast territory and abundant resources. Despite the low per capita resource occupancy and significant contradiction between humans and land caused by a large population base, its vast national territorial area and its rich types and quantities of resources enable its overall ecological environment to have certain

advantages over other countries and regions. On the other hand, China has been in an economic growth mode of high pollution, high energy consumption, and in the environmental protection mode of "pollution first and treatment afterwards" for a long period. This has caused China to have to face enormous pressure in the process of industrial transformation and upgrading and in promoting the coordinated development of social economy. According to our evaluation results, China's economic society score increased progressively, while the ecological environment score fluctuated after 2010, which indicated that China's current social and economic development mode was not conducive to the improvement of ecological environment.

### 3.2. The Development Level of Ecological Civilization in Various Provinces, Autonomous Regions, and Municipalities

Provinces (autonomous regions and municipalities) may have different economic and natural characteristics that may yield different causality directions when constructing ecological civilization [37]. Therefore, it was necessary for us to further study the development level of ecological civilization in various provinces, autonomous regions, and municipalities in China. According to our evaluation results of China's ECI scores, the results in 2004, 2008, and 2012 represent China's ecological civilization development in a typical low level, comparatively low, and intermediate level stage, respectively, and the result in 2016 can represent the current level of China's ecological civilization development. Therefore, we selected 2004, 2008, 2012, and 2016 to calculate the scores of economic society, ecological environment, and ECI of China's provinces, autonomous regions, and municipalities in these years in order to analyze the spatial and temporal pattern of ecological civilization development. The results are shown in Table 5.

**Table 5.** Scores of ecological environment (B1), economic society (B2), and ECI of China's provinces, autonomous regions, and municipalities in 2004, 2008, 2012, and 2016.

| Region | 2004 | | | 2008 | | | 2012 | | | 2016 | | |
|---|---|---|---|---|---|---|---|---|---|---|---|---|
| | B1 | B2 | ECI | B1 | B2 | ECI | B1 | B2 | ECI | B1 | B2 | ECI |
| Beijing | 51.28 | 59.71 | 55.49 | 47.47 | 72.42 | 59.95 | 56.37 | 82.35 | 69.36 | 57.86 | 89.02 | 73.44 |
| Tianjin | 48.27 | 49.28 | 48.78 | 54.02 | 56.95 | 55.49 | 55.59 | 70.22 | 62.91 | 52.88 | 79.06 | 65.97 |
| Hebei | 33.30 | 26.23 | 29.77 | 42.57 | 35.36 | 38.96 | 48.68 | 44.45 | 46.56 | 46.30 | 51.11 | 48.70 |
| Shanxi | 21.91 | 27.16 | 24.53 | 36.32 | 34.19 | 35.25 | 50.13 | 42.10 | 46.11 | 42.67 | 49.59 | 46.13 |
| Inner Mongolia | 37.82 | 20.45 | 29.14 | 48.47 | 28.64 | 38.56 | 59.24 | 41.31 | 50.28 | 57.62 | 48.99 | 53.30 |
| Liaoning | 40.27 | 36.47 | 38.37 | 49.67 | 43.09 | 46.38 | 56.39 | 56.49 | 56.44 | 52.87 | 61.05 | 56.96 |
| Jilin | 48.34 | 31.34 | 39.84 | 50.13 | 40.00 | 45.06 | 58.83 | 47.15 | 52.99 | 61.07 | 53.34 | 57.20 |
| Heilongjiang | 40.48 | 29.23 | 34.85 | 52.98 | 37.91 | 45.44 | 61.86 | 44.18 | 53.02 | 61.45 | 49.79 | 55.62 |
| Shanghai | 33.78 | 60.96 | 47.37 | 42.39 | 67.50 | 54.95 | 45.11 | 76.74 | 60.92 | 56.13 | 83.64 | 69.89 |
| Jiangsu | 39.29 | 36.98 | 38.13 | 45.43 | 49.51 | 47.47 | 47.49 | 62.47 | 54.98 | 53.84 | 70.47 | 62.16 |
| Zhejiang | 49.77 | 42.14 | 45.95 | 54.27 | 53.52 | 53.89 | 68.11 | 64.11 | 66.11 | 64.80 | 75.05 | 69.92 |
| Anhui | 31.06 | 21.22 | 26.14 | 46.36 | 29.29 | 37.83 | 57.16 | 42.89 | 50.02 | 60.03 | 51.08 | 55.55 |
| Fujian | 58.21 | 34.03 | 46.12 | 66.40 | 42.00 | 54.20 | 75.75 | 51.78 | 63.76 | 74.56 | 62.83 | 68.70 |
| Jiangxi | 36.82 | 23.61 | 30.21 | 57.74 | 30.79 | 44.26 | 75.94 | 44.33 | 60.14 | 66.18 | 51.78 | 58.98 |
| Shandong | 34.79 | 33.99 | 34.39 | 52.67 | 43.89 | 48.28 | 59.62 | 54.01 | 56.81 | 50.20 | 61.20 | 55.70 |
| Henan | 36.19 | 25.35 | 30.77 | 44.55 | 35.49 | 40.02 | 49.06 | 43.92 | 46.49 | 46.19 | 52.03 | 49.11 |
| Hubei | 33.32 | 24.79 | 29.06 | 45.19 | 33.14 | 39.16 | 57.50 | 44.72 | 51.11 | 52.04 | 53.27 | 52.66 |
| Hunan | 35.02 | 23.12 | 29.07 | 57.24 | 33.29 | 45.27 | 67.90 | 44.14 | 56.02 | 66.13 | 52.79 | 59.46 |
| Guangdong | 47.46 | 39.37 | 43.41 | 63.78 | 46.52 | 55.15 | 64.91 | 59.44 | 62.17 | 65.37 | 66.93 | 66.15 |
| Guangxi | 42.26 | 19.26 | 30.76 | 55.51 | 26.19 | 40.85 | 70.71 | 38.01 | 54.36 | 70.71 | 45.90 | 58.31 |
| Hainan | 55.65 | 25.23 | 40.44 | 70.13 | 30.47 | 50.30 | 75.13 | 40.69 | 57.91 | 75.31 | 47.76 | 61.53 |
| Chongqing | 40.65 | 25.45 | 33.05 | 52.97 | 33.03 | 43.00 | 66.77 | 46.03 | 56.40 | 65.26 | 55.44 | 60.35 |
| Sichuan | 39.39 | 19.71 | 29.55 | 62.30 | 23.97 | 43.13 | 65.50 | 33.29 | 49.39 | 58.16 | 48.34 | 53.25 |
| Guizhou | 32.47 | 9.19 | 20.83 | 48.93 | 14.15 | 31.54 | 61.60 | 22.56 | 42.08 | 63.21 | 39.62 | 51.41 |
| Yunnan | 52.55 | 12.91 | 32.73 | 67.04 | 17.45 | 42.24 | 68.06 | 25.89 | 46.97 | 70.53 | 40.53 | 55.53 |
| Tibet | 58.01 | 7.97 | 32.99 | 60.06 | 9.99 | 35.02 | 62.68 | 14.13 | 38.40 | 64.16 | 20.20 | 42.18 |
| Shaanxi | 20.45 | 21.37 | 20.91 | 35.61 | 28.97 | 32.29 | 46.93 | 43.72 | 45.33 | 47.49 | 55.42 | 51.45 |
| Gansu | 19.57 | 10.97 | 15.27 | 27.60 | 15.11 | 21.35 | 42.37 | 24.44 | 33.41 | 47.65 | 36.84 | 42.24 |
| Qinghai | 57.16 | 13.05 | 35.10 | 56.81 | 15.81 | 36.31 | 66.93 | 22.70 | 44.81 | 60.98 | 35.86 | 48.42 |
| Ningxia | 28.30 | 19.87 | 24.08 | 36.94 | 22.57 | 29.75 | 46.22 | 29.04 | 37.63 | 47.32 | 38.56 | 42.94 |
| Xinjiang | 41.66 | 19.08 | 30.37 | 46.56 | 22.45 | 34.51 | 53.03 | 30.82 | 41.92 | 53.52 | 34.58 | 44.05 |

In general, there were different development levels of ecological civilization in China's various provinces, autonomous regions, and municipalities. The overall development level of eastern coastal provinces and municipalities was significantly higher than that of central and western regions. With the development level of ecological civilization rising year by year, the spatial equilibrium of ecological civilization development was continuously enhanced. The differences in the development level gradually decreased, but the differences among regions were still apparent.

In 2004, most districts were at a low level or even extremely low level of ecological civilization development (even the southeastern coastal areas, where the development level of ecological civilization was higher, were merely at low levels). Beijing was the only district that had achieved an intermediate level of ecological civilization development. In 2008, most districts were at the low level or comparatively low level of ecological civilization development, the southeastern coastal areas were at the intermediate level. In 2012, China made great progress in developing its level of ecological civilization—the level in most regions improved for one stage compared with 2008; some districts with good resource endowment and rapid economic development even improved for two stages. In 2016, Beijing took the lead in ecological civilization development and reached a high stage, however, the environmental and social problems caused by China's long-term irrational economic development mode became increasingly obvious. The growth of ecological civilization development in China's various districts slowed down significantly, and the economic transformation was imperative for China in order for it to achieve social progress and sustainable economic development.

In terms of the results in Tables 3 and 5, the development level of ecological civilization in various provinces, autonomous regions, and municipalities could be summarized into the following five types [32]:

(1) Highly developed economy type: including Beijing, Tianjin, Shanghai, Jiangsu, Zhejiang, and Guangdong. This type required the ECI scores in districts to be above 60 in 2016, and the score of ecological environment was lower than that of economic society in recent years. These districts were advanced regions in the construction of ecological civilization and had the highest ECI score in China; however, they were merely at a comparatively high level of ecological civilization development until 2016 (except Beijing). Their development was unbalanced, especially in three eastern municipalities. For these districts, environmental quality was the most important bottleneck that restricted the construction of ecological civilization in the future. In order to move towards a higher stage of ecological civilization construction, these districts should further optimize the industrial structure, accelerate further transformation of development modes, and realize further coordination between ecological environment and social economy.

(2) Moderately developed (lagging economic society) type: including Inner Mongolia, Jilin, Anhui, Fujian, Hunan, Chongqing, and Sichuan. This type required the ECI scores in the districts to be above 50 in 2016 and the ecological environment score to be higher than that of the economic society. The development stage of ecological civilization in almost all districts in this stage was above (or at) the average level of China, which indicated that it had notable progress in the field of ecological environmental protection and social economic construction. Despite all this, the gap between ecological environment and economic society continued to narrow, meaning that these districts still developed their economy at the expense of the environment in recent years, such as Inner Mongolia and Sichuan. The advantages of ecological environment in these two districts are becoming more and more inconspicuous. In the future, such districts should vigorously promote industrial transformation, upgrading, and comprehensively enhancing people's living standards on the basis of maintaining fine ecological and environmental quality.

(3)　Moderately developed (lagging ecological environment) type: including Hebei, Shanxi, Liaoning, Shandong, Henan, Hubei, and Shaanxi. This type required the ECI scores in the districts to be above 45 in 2016, and the score difference was not obvious between ecological environment and economic society, or the score of ecological environment was slightly lower than that of economic society. The basic conditions of ecological environment in these districts were not good, however, the economic development mode in these districts was relatively extensive, which meant that the development level of ecological civilization was average and ascended slowly in spite of the faster economic growth. For example, the ecological environment score in Hubei was even lower than that of economic society for the first time in 2016 due to its extensive development mode, while its ECI score still increased slowly. The pressure of ecological civilization construction in these districts was greater than that of the last type (moderately developed (lagging ecological environment) type) because these districts reached similar levels of ecological civilization at the expense of more ecological environment quality. For these districts, measures should be taken in the future such as carrying out comprehensive environmental protection, controlling pollutant discharge, promoting waste recycling and intensive use of resources, and optimizing development structure so as to prevent the aggravation of human–land conflicts and promote the coordinated development of ecological environment and social economy.

(4)　Environmentally-friendly type: including Heilongjiang, Jiangxi, Guangxi, Hainan, Guizhou, Yunnan, Qinghai, and Tibet. This type required the ecological environment scores in the districts to be above 60 and the economic society scores to be under 50 (the economic society scores in individual districts could be above 50 slightly, but the ecological environment scores should have exceeded the economic society scores at least 10 points). These districts had favorable resource endowment conditions and ecological environment foundations, and their ecological environment was ranked among the top in China; however, social and economic development lagged behind, thus, the ECI scores of these districts were not high. Compared with other regions, such districts will have great potential for the future construction of ecological civilization. Full use should be made of local resource advantages. Local ecological benefits should be translated into economic benefits in the way of promoting comprehensive and coordinated sustainable development of social economy instead of sacrificing ecological environment in such districts in the future.

(5)　Low development type: including Gansu, Ningxia, and Xinjiang. This type required the ecological environment scores in the districts to be under 45. The basic conditions of ecological environment in these districts were average, the ecological environment score was not high, and the social economic development lagged far behind. Therefore, these districts were faced with the most severe conditions of poor ecological civilization development. Although the development level of ecological civilization improved year by year, there is still much room for improvement. In the future, construction of ecological civilization in these districts will be the most arduous task. Not only should intensive use and control of various resources be further strengthened, but these districts should also fully absorb technology and development experiences from advanced regions and vigorously improve local economic development.

## 4. Conclusions and Prospects

This study investigated the development level of ecological civilization in mainland China during 2004–2016 using an index system we established. Our results indicated such conclusions: the development level of ecological civilization in mainland China and its provinces, autonomous regions, and municipalities had different levels of improvement. The differences among regions gradually decreased, though the differences were still apparent. The overall development level of the eastern coastal regions was significantly higher than that of the central and western regions. Regional variations in development levels of ecological civilization cannot be ignored in spite of its gradually decreasing tendency. At present, China is in the intermediate stage and is moving to the comparatively high level of ecological civilization development. This stage is the period when China is vigorously promoting

the construction of ecological civilization. It is the period of China's sustained economic growth, and is also the critical period for China to deal with a series of conflicts of human–land relationships and social contradictions. In addition, we must fully realize that China's entering a higher stage of ecological civilization development greatly depends on its ability to improve ecological environment, because its steady social economic development, we predicted, will last over a long period of time. We can even say that it is instability of ecological environment that will lead to some fluctuation or regression of the development level of China's ecological civilization in the future.

Human-induced changes must not go beyond the thresholds of resistance and resilience that characterize local ecosystems, which is the core requirement of ecological civilization construction [38]. China is in the transition stage of economic development and is realizing the transformation from a high-speed growth mode to a high-quality development mode. By now, China is and will remain the biggest developing country in the world. In various aspects, it is developing far behind the developed countries and regions. The idea of ecological civilization arises at time when China's industrialization and urbanization has not been fully completed. China may complete the rest of the task by means of eco-technological changes and realizing a new pattern in developing its ecological civilization to reduce costs in changing development patterns and to gain a head start in the realization of this new ecological pattern of development [39,40]. There is no doubt that China has some problems in its economic development progress, but the Chinese government has put forward the correct direction by promoting ecological civilization construction. China can realize the necessity of green development in top-level designs and long-term strategies after all. The next step for China is to ensure how these policies will be implemented and how these concepts of ecological civilization development will truly become a new driving force of industry innovation and economic growth.

Another question we should discuss is that, theoretically, we hope that the evaluation index system we established can be applied to evaluate ecological civilization development in all regions, however, we must take local realities into consideration when we select indicators if we want to study the development level of China's ecological civilization. For example, the indicator D26 (Proportion of Nationally Designated Eco-Demonstration Region/Eco-County) is characterized by a distinctive China style. This characteristic index helps us to have a deeper understanding of the situation of the local ecological civilization construction mission, and it enables the evaluation index system to be China-specific to some extent. Nevertheless, most of the indicators in our system also apply to other places, which makes our system an effective reference for the evaluation of ecological civilization development in other places.

The limitations of this study are mainly reflected in the following aspects. Although the idea of combining AHP and the entropy weight method in index selection greatly improved the accuracy of the study, several indicators (especially indicators reflecting the policy guarantee system of ecological civilization) were not considered because of their inaccessibility. Moreover, we believe that the index system we established can reflect the situation of ecological civilization development in China and its provinces, autonomous regions, and municipalities comprehensively, scientifically, and accurately, but we also realize that it is impossible for the evaluation results to be consistent with the absolute reality of each place. In future studies, we will analyze the spatial and temporal patterns of ecological civilization development in various provinces, autonomous regions, and municipalities in China, and further discuss the relationship between ecological environment and social economy based on our evaluation results. Certainly, maybe we will revise our evaluation index system if we find better indicators in the future.

**Author Contributions:** X.W. and X.C. designed the study. X.W. wrote the main manuscript text, collected and analyzed the data; X.C. supervised the manuscript and suggested revisions. All authors read and approved the final manuscript.

**Funding:** This research was funded by the National Key R&D Program of China (2018YFC0704700, 2018YFC0704702), the National Natural Science Foundation of China (41471462) and the Fundamental Research Funds for the Central Universities (lzujbky-2017-42, 18LZUJBWTD016).

**Conflicts of Interest:** The authors declare no conflict of interest.

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
