# Peer review of "An Evaluation Index System of China’s Development Level of Ecological Civilization"

_sustainability, doi:10.3390/su11082270_

Round 1

Reviewer 1 Report

Reviewer Report

Article Title: An Evaluation Index System of China's Development Level of Ecological Civilization

I congratulate the authors for their effort in piecing together this important index-related research.

The paper is consistent with MDPI SUSTAINABILITY and fits in nicely with the overall journal scope. The lit. rev., references and methodology are well done. The research idea is good and its significance within China is very novel. The novelty is high and the indice statistical work is sound; the results are very clear. The Abstract is sufficiently done.

Article features: regarding the originality and research findings

Much of the indexing research and equations are sound. The results show promise and thus the paper attains my approval for publication.

Recommendation: I would recommend this article for publication pending the following MINOR REVISION.

In the discussion or conclusion (where ever the authors feel it would best fit), they should incorporate a paragraph or two to explain why this index was used? why it is relevant for this application? and explain why it is better than other indices that already exist?

Also, explain if the calculative methodology is application-specific to this example (i.e., to China) or not. If not, explain its reciprocity to other places.

Finally, to complete the revision work, incorporate a comparative elucidation of other important indices and potentially the reason why we use indices.

(The answer being: mostly to compare economic-only measurements (like GDP, etc.) to include ecological and social aspects.)

Author Response

Point 1: In the discussion or conclusion (where ever the authors feel it would best fit), they should incorporate a paragraph or two to explain why this index was used? why it is relevant for this application? and explain why it is better than other indices that already exist?  

Response 1: We added two paragraphs after the table 1 (page4-5) , mainly expounds why we choose these indicators when we establish our evaluation index system, and why the indicators we selected are more scientific, comprehensive and accurate, and why the system we established can better reflect the actual situation of local ecological civilization development compared with other studies. We discussed the relationship between ecological environment and economic society in the construction of ecological civilization and explain it with some representative indicators. We put this section here rather than in the conclusion section mainly because we consider that this section is directly related to table 1, and it may be not easy for some readers to understand it if this section is included in the conclusion.

Point 2: Also, explain if the calculative methodology is application-specific to this example (i.e., to China) or not. If not, explain its reciprocity to other places.

Response 2: We added two paragraphs at the end of manuscript (in penultimate paragraph of the revised manuscript) to explain if our calculative methodology is application-specific to China. We think that if we want to study the development level of an area's ecological civilization, we should use some characteristic indexes to take local realities into consideration, which makes the evaluation index system be application-specific to China to some extent. Nevertheless, most of the indicators in our system also apply to other places, which is the way we want it to be.

Point 3: Finally, to complete the revision work, incorporate a comparative elucidation of other important indices and potentially the reason why we use indices.

(The answer being: mostly to compare economic-only measurements (like GDP, etc.) to include ecological and social aspects.)

Response 3: Considering the reason why we use these indices is also one of its advantages, and we believe that this part of the discussion is relevant to the revision for point 1, so we put this part together with them. We also explain how some representative indicators we selected could make up for the shortcomings of the economic-only measurements.

Reviewer 2 Report

The article is well structured and is an original and interesting research. The introduction allows a good understanding of the context and provides updated bibliographical references.

There are a couple of questions to solve:

-   On page 9, 5 groups of cities are differentiated, considering the results of table 5. It should be explained which the exact cut points for each group are, and why these criteria have been followed (if possible, supported by the literature ).

 -   There are no limitations to this work. Some limitations should be included in the conclusions section, which will serve as a basis for the proposal for future research.

Author Response

Point 1: On page 9 (manuscript) , 5 groups of cities are differentiated, considering the results of table 5. It should be explained which the exact cut points for each group are, and why these criteria have been followed (if possible, supported by the literature ).
 Response 1: We revised this part (page 9-10). We added the exact cut points in the section of explaining the five groups of provinces. These criteria are combined with table 3. Our classification of the stages of ecological civilization is based on our own findings, because we have not found relevant literature to provide an exact basis for classification, even what stages should ecological civilization be divided into is still under academic exploration. The reference No.32 we add provides a way to divide the stages of ecological civilization, but the research focus of this article is different from ours. Point 2: There are no limitations to this work. Some limitations should be included in the conclusions section, which will serve as a basis for the proposal for future research. Response 2: We added one paragraphs at the end of the text (in the last paragraph of the revised manuscript) to explain the limitations and possible future research prospects of our study. Due to the addition of this section, title 4 is amended to “Conclusions and prospects”.
